published: 01 July 2021

# Professional Characteristics of Health Promotion: A Scoping Review of the German and International Literature

Verena Biehl[1,2]*, Thomas Gerlinger[2] and Frank Wieber[1,3]

[1]School of Health Professions, Institute of Health Sciences, Zurich University of Applied Sciences, Winterthur, Switzerland, [2]Department of Health Care Systems, Health Policy and Sociology of Health, Faculty of Health Sciences, University of Bielefeld, Bielefeld, Germany, [3]Department of Psychology, University of Konstanz, Konstanz, Germany

**Objective:** This scoping review investigates current developments in the professional characteristics of health promotion (HP) with a focus on the German speaking part of Europe. The conceptualization of HP is a prerequisite for progressing HP professionalization and clarifying the interconnectedness between HP and Public Health.

**Methods:** The search strategy was informed by sociological professionalization theories. Original publications were included in a content-based analysis.

**Results:** Ninety publications (37 original publications) were identified in the review. The results are summarized in categories based on professional characteristics: 1) profession, 2) ethics, 3) education/training 4) competencies, and 5) quality. The professionalization of HP regarding the professional characteristics is less developed in the German compared to the international literature.

**Conclusion:** The mixed findings emphasize the relevance of a common HP conceptualization. The HP core competencies, which have been developed by the International Union for Health Promotion and Education must be further promoted. A strong HP workforce within Public Health strengthens the HP status in policy contexts and society and its contribution to promoting health and tackling social inequalities in health.

**Keywords:** health promotion, public health, professionalization, professional characteristics, professional competencies

**Edited by:**
Alberto Borraccino,
University of Turin, Italy

**Reviewed by:**
Dean Whitehead,
University of Tasmania, Australia
Helen Keleher,
Monash University, Australia

**\*Correspondence:**
Verena Biehl
verena.biehl@zhaw.ch

**Citation:**

## INTRODUCTION

In pursuing the overarching goal of preventing and managing health problems worldwide, health promotion (HP) offers valuable competencies that can contribute to political and societal efforts. With its focus on socioenvironmental determinants of health at a community level as described in the Ottawa Charta in 1986, HP has the potential to promote health and well-being for all groups of society [1,2]. In line with this increasing recognition of the relevance of HP, the United Nations explicitly referred to it in the Sustainable Development Goals for 2030 in goal 3: "Enable healthy lives and promote well-being for all at all ages" [3]. These goals are particularly important given that individualism, urbanization and globalization are driving many major contemporary health issues, including noncommunicable diseases, and call for rethinking our health systems with a strengthened focus on HP and prevention that complements the traditional focus on disease management and health care [4,5]. However, the often suboptimal quality of HP practice [6–8] and the lack of Public

Health (PH) professionals [9,10] call for a stronger HP workforce. Although HP has come a long way since the 1980s, it is not well-established as a profession in our society yet, at least in the German speaking part of Europe [11–15].

For better comprehensibility the following section provides descriptions for the central terms in this article and their relation between each other referring to Mieg, 2016 [16]. Professionalization in this article is understood as an overall term for the development of the field of action of HP being performed by professionals. Within the professionalization process a main requirement is the conceptualization of the concrete field of action, which can then be described by specific professional characteristics. As identification of professional characteristics therefore represents a central building block to support the development of the HP profession, this study aims to provide an overview of the professional characteristics of the HP profession described in the literature. Thereby, professional characteristics don't capture the analysis of a whole profession but facilitate the understanding of different relevant aspects of a single profession. We understand professional HP as it was conceptualized in the Ottawa Charta: a salutogenic perspective on health focusing on populations based on the principles of participation, empowerment and health equity. HP is a multilayered concept to initiate a paradigm shift within our health systems to a state of Health in all Policies [1]. This means many professions are involved in this paradigm shift including a specific profession - the HP practitioner.

Professionals are needed to face complex challenges in today's knowledge-oriented societies, such as the promotion of well-being and population health. As systematic approaches to describe professional characteristics, a variety of sociological theories on professionalization have been developed since the early 20th century. For instance, professional characteristic theories represent pragmatic approaches to the issue of professionalization that focus mainly on the institutional level, namely on education and training institutions, professional institutions or a code of ethics [17–19]. Since the 1980s, however, the focus shifted to the individual level of professions, such as the professional performance [20], competencies and professional identity [16]. A combination of different approaches that distinguishes the institutional level and the individual level of professionalization was then suggested by Mieg [16]. Summarizing these different professional characteristics on institutional and individual level [16–20] leads to the following shortened listing: 1) professional performance/professional identity 2) ethics/values, 3) education/training, and 4) quality/standards/competencies. These professional characteristics are a helpful conceptual frame for analyzing the conceptualization of the HP profession and they are thus used to derive the research questions of this scoping review, which aims to provide an overview of current developments regarding the professional characteristics of HP.

While HP is clearly a central part of PH, it is important to consider how the two are connected [21,22]. As in HP, the professionalization discussion is also ongoing for PH [10,23–25]. PH deals with a great variety of issues–including prevention, health care and tackling the social determinants of health [21,26] (e.g., see 10 Essential Public Health Operations;

[27]) making PH very broad and complex to capture [10,26,28–30]. Moreover, PH is described as research orientated [28,31], and still overlaid by biomedical principles [32,33] whereas HP is clearly practically orientated to address the socioenvironmental determinants of health at a community level with a bio-psycho-social perspective. PH has to recognize the necessity of specific skills for HP [34]. There are international efforts to foster the professionalization of HP that are illustrated by several indicators found in literature [2,14,21,22,35–37] and shortly listed in **Table 1**. Although the specialization of competencies in HP is a desirable outcome to tackle todays' pressing health problems to foster quality standard in HP practice, a gradual separation between PH and HP represents a major challenge for HP and PH because the workforce capacity of each discipline may be reduced by a segregation of these interconnected disciplines. The professional HP development within PH is therefore a great concern to the workforce, which needs to be closely monitored in order to get the best of both worlds: a clear profile for HP professionals and a synergistic interplay of HP and PH. We argue that identifying the professional characteristics of HP in the literature helps to foster the conceptualization of HP which contributes to the professionalization and is a prerequisite for analyzing and optimizing the interconnectedness between the HP and PH professions. The aim of this study is therefore to provide an overview of the professional characteristics of the HP profession focusing on German compared to international literature.

## METHODS

Literature on HP as a profession, especially empirical studies is scarce [11,15]. Therefore, we conducted a scoping review as it allows the inclusion of literature of lower scientific quality such as grey literature, and is well suited to gain a broad and explorative understanding of the research question and can even help to specify it [38,39].

As outlined in the introduction, a summary of different sociological professionalization theories referring to professional characteristics on the institutional and individual level [16–20] provided the conceptual background for the research questions of this scoping review. Four categories in which the professional characteristics are gathered were differentiated: 1) Professional performance/professional identity, 2) ethics/values, 3) education/training, and 4) quality/standards/competencies. Building on the four distinct categories of the different sociological professionalization theories [16–20] described in the introduction we derived the following focal research questions:

"What does the literature reveal about . . .

1) . . . the profession of HP, with a focus on professional identity and professional performance?"
2) . . . professional ethics/values of HP?"
3) . . . education and training in HP?"
4) . . . quality/standards or competencies of HP?"

**TABLE 1 |** International efforts to foster the professionalization of health promotion that are illustrated by several indicators found in literature [2, 14, 21, 22, 35–37] (Professional characteristics of health promotion: a scoping review of German and international literature, Switzerland, 2021).

**Indicators for professionalizing HP[a] internationally**

1) *Education and training programs* in HP are established on an international level comprising bachelor's and master's degree programs and continuing education
2) *Core competencies* are defined in a HP framework on an international level [37]
3) *Research units* specialize in HP (e.g. at the Universities of Ireland and Bielefeld) and specific scientific journals are published (e.g. Health Promotion International, Global Health Promotion, Prävention und Gesundheitsförderung)
4) *National and international conferences* on HP are held (e.g. World Conference on HP)
5) *National and international professional institutions* are established, especially the International Union for Health Promotion and Education (IUHPE)
6) *Accreditation systems* are available and applied, such as the "IUHPE Registered Health Promotion Practitioner"
7) *National and international political priorities* are established on HP (e.g. UN Sustainable Development Goals (SDG 3) or the German law on prevention and HP)

*[a]HP: health promotion.*

**TABLE 2 |** Search strategy elaborated with PICO-scheme including search terms, synonyms and Boolean operators (Professional characteristics of health promotion: a scoping review of German and international literature, Switzerland, 2021).

| | Boolean operators | Search terms and synonyms |
|---|---|---|
| **P** (opulation) | AND | Health Promotion |
| **I** (ndicator) | | *not relevant for this search strategy* |
| **C** (omparison) | NOT | Public Health |
| **O** (utcome) | AND | professional identity OR collective identity OR vocational identity OR occupational identity OR identity OR identification OR profession[a] OR ethic[a] OR value OR norm OR behavior OR standard OR competenc[a] OR frame[a], OR quality OR profile OR challenges OR education OR study OR training OR university |

*[a]: truncations are used to allow more search results for specific keywords*

In order to systemize the search strategy, we applied the PICO scheme: P (opulation), I (ndicator), C (omparison), O (utcome of interest) [39]. Although, not all categories were relevant for the present research question, the scheme helped to specify the search strategy. All keywords, synonyms and the Boolean operators that have been used are shown in **Table 2**.

The search strategy employed three databases: CinahlComplete, PubMed and the International Bibliography of the Social Science (IBSS). The strategy was adapted to the search template in the database. Moreover, specific journals were screened (Global Health Promotion, Health Promotion International, Prävention und Gesundheitsförderung (German) and Spectra (Swiss) and Google Scholar was searched for grey literature. We also conducted a hand search in the references of relevant literature. The scoping review was conducted in March 2020.

Study selection was guided by inclusion criteria. We included 1) any kind of theoretical and original (empirical studies and literature reviews) literature, including journal articles and grey literature; 2) literature that focuses on HP as defined in the Ottawa Charta [1]; and 3) German as well as English literature in the review. The German literature represented a special interest of the authors who plan to conduct further research in the German speaking part of Europe (Switzerland). The inclusion of international literature supports the identification of potential opportunities and challenges within the conceptualization of the HP profession as many countries have advanced further in the professionalization of HP (e.g. Australia, New Zealand, Canada and Ireland) [6,22]. Finally, we D) limited the publication date to 2012 to 2020 as we expected the standardized competencies that were published by the IUHPE in 2011 in the CompHP core competencies framework of HP (37; German version in 2014 [40]) to impact the professionalization

debate in HP. **Figure 1** shows the search process including all literature identified ($N = 1880$), those who were rated as relevant after screening the abstracts ($n = 119$) and – in a second screening – the full texts ($n = 90$). For the content analysis, we selected original literature only ($n = 37$).

The data was collected in an Excel file containing relevant information: author, nature of literature (journal article, grey literature), year of publication, title, methods, aim, outcome and country study conducted (see **Table 3**). Data analysis was based on deductive and inductive principles. Within the deductive part, we categorized the data comparable to the professional characteristics as a theoretical basis of the search strategy. The data is described quantitatively using frequencies (see **Table 4**). For the inductive data analysis, we conducted a content-based literature analysis, which was only applied to original literature ($n = 37$) (see **Table 3**).

## RESULTS

The search strategy revealed 90 publications, of which 31 are in German language and 59 in English of which non was conducted in a German speaking region. In total, 37 original publications (empirical studies and literature reviews) published between 2012 and 2020 were identified, 24 of which were in English. An overall lack of empirical studies on the HP profession was noteworthily, especially in the German speaking countries but also internationally. **Table 4** provides further details about the descriptive analysis of the scoping review results of all 90 publications.

The content-based analyses revealed five major categories: 1) profession, 2) ethics, 3) education/training, 4) competencies and 5) quality. As the focus and content of the literature identified in the

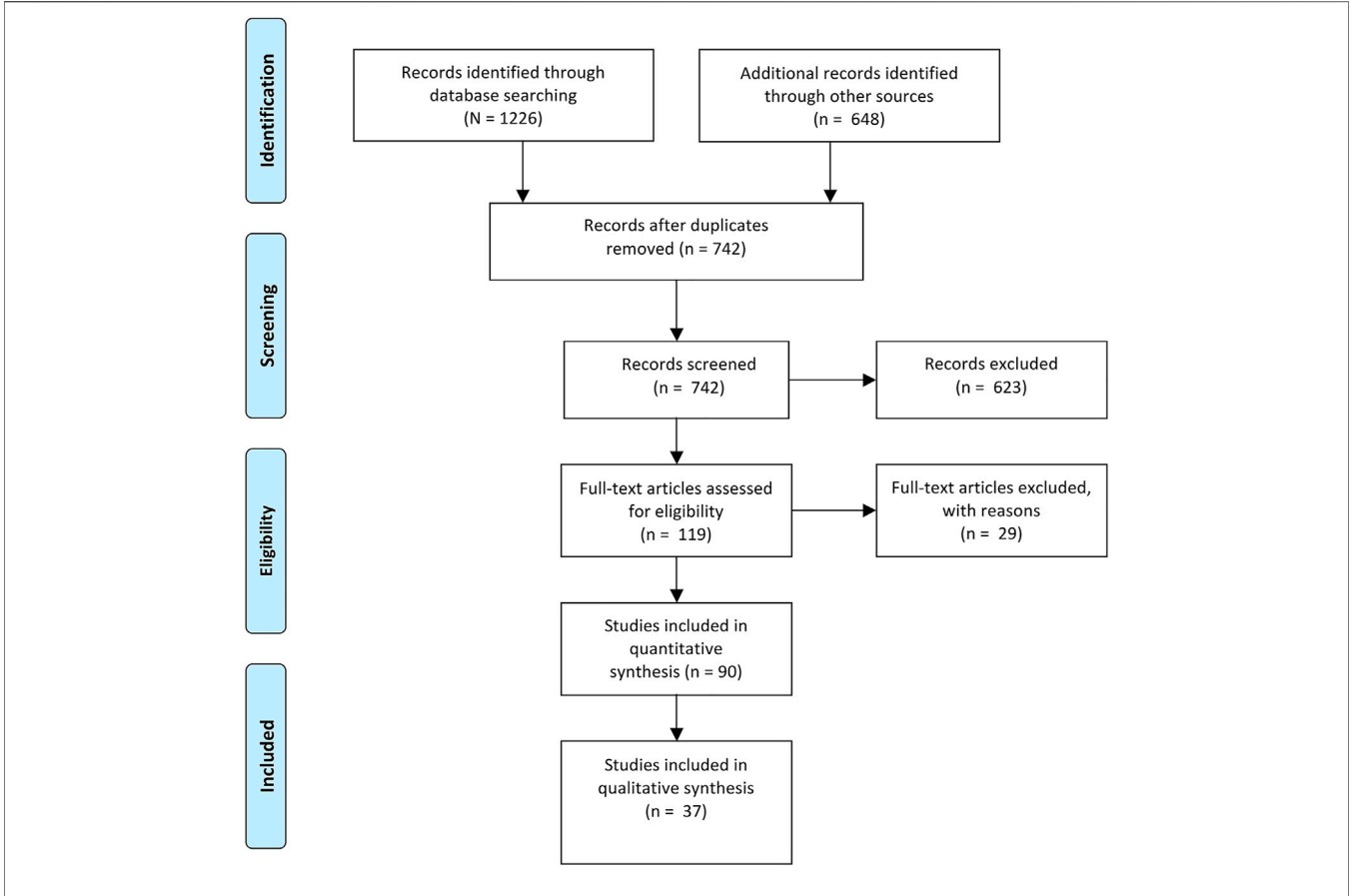

**FIGURE 1** | Flowchart of literature included in the scoping review. (Professional characteristics of health promotion: a scoping review of German and international literature, Switzerland, 2021)

scoping review partly differed from the professional characteristics that have been outlined in the sociological theories, we specified and adapted the names of the categories as follows: 1) "Profession" summarizes literature focusing on HP as a profession including professional performance and professional identity; 2) "Ethics" summarizes literature regarding ethics and values of the HP profession; 3) "Education/training" summarizes literature describing training programs and study courses in HP; 4) "Competencies" was separated from 5) "Quality" as a category because there were numerous publications found which either dealt with competencies of HP professionals–or with analyzing quality standards of HP practice. The content of the original publications is summarized in the following section (see **Table 3** for more details of all 37 original publications).

## Profession
Three publications in German language discuss the HP profession, focusing on education of HP [41], employees' perspectives on the profession [14] and an analysis of decision makers of HP interventions in Germany [42]. Overall, the study's results show unclear definitions and roles of HP practitioners, especially when practitioners are not having an education or training in HP or a related field [14,41]. The variety of existing health related study

courses in Germany add to the intransparency of competencies and curricula [14]. These barriers hinder the development of a professional identity and the societal acceptance of the profession [14]. Compared to the German literature, the English literature (mainly Australia, Canada and United States) seems to fulfill higher quality standards as they mostly consist of peer-reviewed journal articles. Also, with respect to the content, English articles focus on specific aspects of professional performance in HP, such as critical reflection [43], sustainable practice [44,45] and on socioecological determinants in interventions [8]. However, English and German literature are both clear about the importance of specifically trained HP competencies [8,14,42,45]. These publications variously emphasized the value of a code of ethics, a meta theory in HP and a common professional profile including professional competencies that would help to promote professionalization of HP regarding professional identity and professional performance in HP.

## Ethics
All six original publications identified are published in English (mostly Australia) and focus on the ethical aspects of HP practice. Some publications focus on defining ethical dimensions of HP [46,47], one focuses on challenges of empowerment applied in

**TABLE 3 |** Data extraction of original publications identified in the scoping review chronologically ordered by category and language published (Professional characteristics of health promotion: a scoping review of German and international literature, Switzerland, 2021).

| Authors (nature of literature) | Title | Year | Method | Aim | Outcome | Country |
|---|---|---|---|---|---|---|
| **Category: Profession** | | | | | | |
| Zocher, U. (grey literature) | About the challenge to study and teach health promotion–results of participatory evaluation | 2013 | Mixed methods evaluation study | Evaluating student's perspective on the profession of HP[a] including strength and challenges of the study course | The holistic approach of HP for some students is hard to capture. Problematic are the diverse professional backgrounds of lecturers. Main conclusions for professionalization in HP are: 1) Expert knowledge/professional understanding (theory on HP), 2) Didactic for HP (reflective processes), 3) Leadership culture in sense of HP | Germany (g)[b] |
| Pawlis, S., Boettcher, A., Metzner, F., Plaumann, M. and Walter, U. (journal article) | Relevance and implementation of prevention in Germany–a national survey of decision makers in health insurance association, state and communities | 2015 | Quantitative online survey | Identifying the relevance and implementation of HP and prevention at decision makers' level at health insurance companies, the state and communities | Intervention of health insurances are mainly pathogenetic oriented, whereas interventions of the state are also focusing on social inequities. Policy makers are more positive towards capacity building in HP and prevention than health insurance companies. Main barriers are financial issues and personal resources as well as unclear roles. | Germany (g) |
| Walter, S. (master thesis) | HP on the way to a profession–an interview study | 2015 | Qualitative study using interviews | Exploring the field of action and the need of HP practitioners asking experts in the field | HP is currently performed by a variety of professions. Most organizations employ at least one person with a professional background in health sciences, whereas related disciplines (sports, nutrition) don't see a gain for the field of action. There are also persons working in HP who don't have any specific or related training. There was agreement on the need of professionals in HP, but the diversity of study programs in Germany related to health are seen as a barrier, lacking transparent competencies. Unclear definitions of roles in HP are barriers for developing a professional identity. The study revealed clear chances of professionalization of HP, but some major challenges which seem to be solvable. | Germany (g) |
| Golden, S. D. and Earp, J. A. (journal article) | Social ecological approaches to individuals and their contexts: twenty years of Health Education and Behavior health promotion interventions | 2012 | Literature review | Identifying socio-ecological approaches in HP interventions published in the journal "health education and behavior" between 1989 and 2008 | Out of 157 articles, 2/3 focused on the individual and not specially on institutions, communities or politics. Interventions in settings like schools or the community or with topics such as nutrition and physical activity were more often integrating socio-ecological approaches compared to other topics. Competencies in HP have to be strengthened in order to address socio-ecological determinants of health in interventions. | United States (e)[c] |

*(Continued on following page)*

**TABLE 3 |** (*Continued*) Data extraction of original publications identified in the scoping review chronologically ordered by category and language published (Professional characteristics of health promotion: a scoping review of German and international literature, Switzerland, 2021).

| Authors (nature of literature) | Title | Year | Method | Aim | Outcome | Country |
|---|---|---|---|---|---|---|
| Harris, N. and Maria, S. (journal article) | Defining sustainable practice in community-based health promotion: A Delphi study of practitioner perspectives | 2013 | Quantitative Delphi study | Identifying definitions and features of sustainable practice in community-based HP practice | Definitions of sustainable practice highlight the importance of collaboration, health determinants, aspirations, processes and outcomes. Four specific features were identified: 1) effective relationships and partnerships, 2) evidence-based decision making and practice, 3) emphasis on building community capacity, 4) supportive contexts | Australia (e) |
| Tretheway, R., Taylor, J., O'Hara, L. and Percival, N. (journal article) | A missing ethical competency? A review of critical reflection in health promotion | 2015 | Narrative literature review | Exploring the current use of critical reflection in HP | Very limited literature exists on critical reflection in HP, whereas its potential to support critical practice is widely acknowledged. Critical reflection is seen as a core competency for HP practitioners to reflect the ethical issues of HP practice. | Australia (e) |
| McFarlane, K., Judd, J., Devine, S. and Watt, K. (journal article) | Reorientation of health services: enablers and barriers faced by organisations when increasing health promotion capacity | 2016 | Systematic literature review | Identifying enablers and barriers of primary health care organizations integrating HP approaches and capacity building of HP | 25 articles were identified which summarize enablers for primary health care organizations: Management support, skilled staff, committed staffing and financial resources, leadership and external partners to collaborate; barriers were the opposite of the enablers including competing priorities to HP within a health care organization which is quite often the case. | Australia (e) |
| Legrand, K., Minary, L. and Briançon, S. (journal article) | Exploration of the experiences, practices and needs of health promotion professionals when evaluating their interventions and programs | 2018 | Quantitative written survey | Describing practices of HP practitioners regarding evaluation of their interventions and identifying the need of an evaluation and monitoring tool | More than half of 246 respondents in total had difficulties with conducting process evaluation. On the one hand selecting the proper evaluation tool was difficult and on the other hand a lack of time and finances. Authors identify a need for an online tool for evaluation of HP interventions. Evaluation is a major competence in HP practice. | France (e) |

**Category: Ethics**

| Authors (nature of literature) | Title | Year | Method | Aim | Outcome | Country |
|---|---|---|---|---|---|---|
| Carter, S. M., Klinner, C., Kerridge, I., Rychetnik, L. and Li, V. (journal article) | The ethical commitment of health promotion practitioners: an empirical study from New South Wales, Australia | 2012 | Qualitative study using interviews | Describing the good in HP by conducting interviews and observations to complete the theory of HP ethics | Ethics in HP has substantive and procedural dimensions: substantive: meliorism, holism, setting approach, focus on primary prevention and communities. Procedural: interventions built on respectful relationships, flexible collaboration with communities, and are capabilities oriented and sustainable. | Australia (e) |

**TABLE 3 |** (*Continued*) Data extraction of original publications identified in the scoping review chronologically ordered by category and language published (Professional characteristics of health promotion: a scoping review of German and international literature, Switzerland, 2021).

| Authors (nature of literature) | Title | Year | Method | Aim | Outcome | Country |
|---|---|---|---|---|---|---|
| Bull, T., Riggs, E. and Nchogu, S. N. (journal article) | Does health promotion need a code of ethics? Results from an IUHPE mixed methods survey | 2012 | Online survey including closed and open-ended questions | Identifying the global need of a code of ethics in HP | Low response rate (11.8%, $N$ = 236). 83% confirmed the need of a code of ethics to clarify roles and definitions of HP. Main ethical issues are: equity, social justice and respect. The CompHP was seen as a possible basis for developing a code of ethics. | International IUHPE (e) |
| Vallgarda, S. (journal article) | Ethics, equality and evidence in health promotion - Danish guidelines for municipalities | 2014 | Documental analysis | Analyzing the content of Danish national guidelines for HP regarding equity, evidence and ethical aspects | Results show a low integration of equity, evidence and ethics in the national guidelines for HP, even though it was declared as an aim of the Danish national board of health. To meet this aims in practice of HP, more effort of public health authorities is necessary. | Denmark (e) |
| Spencer, G. (journal article) | "Troubling" moments in health promotion: Unpacking the ethics of empowerment | 2015 | Ethnographical mixed methods study | Critical analyses of the outcomes of empowerment in HP interventions, exemplified with adolescence health | Young adults' health priorities are set differently, e.g. smoking and drinking promotes their quality of life and self-determination. The different priorities raise some ethical dilemmas for HP practice regarding using empowerment. This reinforces the importance of ethical reflection of HP, especially the dimension of empowerment. | Australia (e) |
| Axford, A. and Carter, D. (journal article) | Building workforce capacity for ethical reflection in health promotion: a practitioner's experience | 2015 | Case study | Analyzing the effect of a program for capacity building in HP on the influence of ethical reflection in practice | The implemented program on capacity building, which consisted of different components based on organizational change management, was evaluated by participants to have a positive impact on ethical reflection in daily practice. The ethical framework was seen as a very practical tool. Challenges for capacity building in this rural area were different professional backgrounds of HP practitioners. | Australia (e) |
| Reilly, T., Crawford, G., Lobo, R., Leavy, J. and Jancey, J. (journal article) | Ethics and health promotion practice: exploring attitudes and practices in Western Australian health organizations | 2016 | Qualitative study using interviews | Analyzing attitudes, practices, enablers and barriers of applied ethics in HP practice | Out of ten interviews, most practitioners include ethical principals in their practice. Enablers were: support with ethical applications, tools and training in ethical practice, organizational support. Barriers were: limited time resources, little knowledge, ethical application not noticed as one's task. HP practice is still lacking evidence-based practice and including ethical principles, which must be amended in the future. | Australia (e) |

**TABLE 3 |** (*Continued*) Data extraction of original publications identified in the scoping review chronologically ordered by category and language published (Professional characteristics of health promotion: a scoping review of German and international literature, Switzerland, 2021).

| Authors (nature of literature) | Title | Year | Method | Aim | Outcome | Country |
|---|---|---|---|---|---|---|
| **Category: Education/training** | | | | | | |
| Sachs, I. and University of Applied Sciences Neubrandenburg (grey literature) | Study courses and job perspectives of graduates in nursing and health sciences | 2014 | Quantitative online survey | Identifying: Conditions of study program, progress, competencies, follow up, job search, job description, job satisfaction | Participation of five study programs: BA health science results: 66% prior professional education (BA), 56% satisfied, 15% dissatisfied with study program, 2/3 direct access to job, 1/3 master program, high satisfaction with job situation, broad field of actions (few in HP and prevention) | Germany (g) |
| Hartmann, T., Baumgarten, K., Dadacynski, K. and Stolze, N. (journal article) | Public health and health promotion in Germany. State of development of bachelor's and master's courses | 2015 | Online search in data bases for health-related study programs in Germany | Overview of accredited study courses in public health and HP in Germany | 43 courses identified, 13 bachelor's and 30 master's at universities and universities of applied sciences, eight bachelor's, five master's in HP, great effort is necessary to synchronize the curriculum content | Germany (g) |
| Lang, G. and Gajar, P. (journal article) | Continuing education and training in health promotion. A survey about intentions and conditions among participants of a countrywide seminar program in Austria | 2018 | Quantitative online survey | Identifying factors influencing the intention to participate in continuing education and training in HP | Influencing factors identified: Self-efficacy, attitudes toward seminars, social expectations of significant others, seminar attendance in the past, interplay of personal behavior and institutional conditions | Austria (g) |
| Tremblay, M. C., Richard, L., Brousselle, A. and Beaudet, N. (journal article) | How can both the intervention and its evaluation fulfill health promotion principles? An example from a professional development program | 2013 | Case study | Describing core principles of HP, the professional development program and its evaluation (HP Laboratory) | The HP Laboratory is a program to foster HP competencies in public health professionals. Participants with different professional backgrounds working in health departments developed a common understanding of HP core principles. Evaluation of the program on HP principles was challenging. | Canada (e) |
| Tremblay, M. C., Richard, L., Brousselle, A. and Beaudet, N. (journal article) | Learning reflexivity from a health promotion professional development program in Canada | 2014 | Qualitative study using interviews | Describing outcomes of reflexivity of the professional development program in HP. Implications of reflexivity for HP practice | Formal and critical reflection are important methods in HP practice and helped the participants to understand their professional roles. Especially critical reflection was hard to encourage, but is important for a critical societal perspective on social justice and equity concerns in HP. | Canada (e) |
| Wilkins, A., Lobo, R. C., Griffin, D. M. and Woods, H. A. (journal article) | Evaluation of health promotion training for Western Australian Aboriginal maternal and child health sector | 2015 | Quantitative evaluation study (online survey or telephone interview) | Evaluation of a HP training for health professionals - impact of using information given in the training - barriers of implementing the trainings' resources in practice | Low response rate (N = 17); diverse professional backgrounds of participants; Job descriptions often with HP, but no training in HP; Chances of the training: planning tools for HP; networking in HP; Barriers: lacking finances, low organizational support | Australia (e) |
| Wiggins, N. and Pérez, A. (journal article) | Using popular education with health promotion students in the United States | 2017 | Case study using mixed methods | Evaluation of a course in popular education method within a master's in public health. This method is supposed to train skills for a systemic analysis of power and privilege to address health and social inequities. | The method was very useful for students to learn focusing attention and creating community, cooperative learning to support accountability to one another and dramatic techniques. Barriers for using this method are students who are not used to liberal pedagogy | United States (e) |

**TABLE 3 |** (*Continued*) Data extraction of original publications identified in the scoping review chronologically ordered by category and language published (Professional characteristics of health promotion: a scoping review of German and international literature, Switzerland, 2021).

| Authors (nature of literature) | Title | Year | Method | Aim | Outcome | Country |
|---|---|---|---|---|---|---|
| Torres, S., Richard, L., Guichard, A., Chicchio, F., Litvak, E. and Beaudet, N. (journal article) | Professional development programs in health promotion: Tools and processes to favor new practices | 2017 | Case study using mixed methods | Evaluation of a professional development program which focuses on improving HP interventions including health promoting principles | and controversy environment of the university. The results showed positive findings implementing several principles of HP (equity, holism, an ecological approach, intersectorality and sustainability). Participants had problems integrating empowerment and participation in their interventions developed. | Canada (e) |
| Komro, K. A., Lang, D. L., Reisinger Walker, E. and Harper, P. D. (editorial) | Integrating structural determinants into MPH training of health promotion professionals | 2018 | Review of curricula of master's in public health | Identifying structural and social determinants in curricula of master's in public health in the United States | 16% of 275 master's in public health contain structural determinants of health in the curriculum: Study courses in social science focus more on social and structural determinants of health; interprofessional collaboration is necessary to efficiently address these determinants of health | United States (e) |

**Category: Competencies**

| Authors (nature of literature) | Title | Year | Method | Aim | Outcome | Country |
|---|---|---|---|---|---|---|
| Baumgarten, K., Blättner, B., Dadaczynski, K. and Hartmann, T. (journal article) | The German professional qualification framework for bachelor's and master's degrees in health sciences/public health and health promotion | 2015 | Framework development based on content analysis | Developing a qualification framework for health sciences/public health and HP bachelor's and master's programs to enable comparability | Involving nine universities a shared framework was developed for bachelor's programs based on the Public Health Action Cycle and Dublin descriptors. No consensus was found for the heterogenic master's programs. | Germany (g) |
| Karg, S., Blättner, B., Krüger, K. and Micheew, N. (journal article) | Competences for working in health promotion. Perceptions of stakeholders | 2020 | Interview study | Identifying stakeholders' needs of competencies in HP and possibilities for students preparing for employment during study course | There was agreement on professional competencies: project management, networking and teamwork, conceptual work, research, public communication, as well as social competencies. Stakeholders appreciate the holistic approach of HP practitioners. | Germany (g) |
| Speller, V., Parish, R., Davison and Zilnyk, A. (journal article) | Developing consensus on the CompHP professional standards for health promotion in Europe | 2012 | Mixed methods study | Description of the process of framework development (CompHP) including testing for acceptance at pan European level | There was a great acceptance and agreement upon the framework CompHP. This goes along with some concerns: 1) high level for practitioners, 2) developed for master's programs, 3) relationship between Public Health and HP | United Kingdom (e) |
| Madsen, W. and Bell, T. (journal article) | Using health promotion competencies for curriculum development in higher education | 2020 | Case study | Analyzing chances and challenges of one institution by developing undergraduate and postgraduate courses building on national competencies of HP. | Main chances of implementing national competencies in curricula of HP education are transparency and comparability. Challenging were organizational structures, characteristics of the study cohorts and capacity of lecturers as well as national and international expectations. One main risk is the lacking flexibility of the curriculum. This was fully obvious when CompHP was developed internationally in the meantime. The institution could restructure the curriculum again on CompHP. | Australia (e) |

**TABLE 3 |** (*Continued*) Data extraction of original publications identified in the scoping review chronologically ordered by category and language published (Professional characteristics of health promotion: a scoping review of German and international literature, Switzerland, 2021).

| Authors (nature of literature) | Title | Year | Method | Aim | Outcome | Country |
|---|---|---|---|---|---|---|
| Battel-Kirk, B., Van der Zanden, G., Schipperen, M. Contu, P., Gallardo, C., and Barry, M. M. (journal article) | Developing a competency-based pan-european accreditation framework for health promotion | 2012 | Mixed methods study including focus groups, online survey and web-based consultation | Find consensus on CompHP pan-European accreditation framework for HP practice, education and training | 405 participants out of 29 countries were involved in the study. Mainly positive agreement on an accreditation framework to assure quality and competence in HP. Barriers for implementation were mentioned regarding lacking resources for implementation of the accreditation framework. Furthermore, the unclear interrelationship between Public Health and HP are barriers for implementation in some countries. | Europe (e) |
| Hicks, K. (grey literature) | Health promotion competencies baseline implementation survey report | 2013 | Quantitative written survey | Collecting information on the current knowledge and implementation of New Zealands' HP competencies amongst the HP workforce | 105 responses revealed that 88% had at least some knowledge on the national HP competencies. The competencies are used for personal development plans, performance reviews, planning programs. They realized an increase of clarity of their professional role and understanding of HP. Challenges for implementing the competencies were other organizational priorities. | New Zealand (e) |
| Battel-Kirk, B. and Barry, M.M. (journal article) | Has the development of health promotion competencies made a difference? A scoping review of the literature | 2019 | Scoping review | Explore current impact of HP competencies on practice, education and training in Europe | 39 sources were identified, mainly focusing on competency frameworks and their development, some report on the use of the frameworks and only two evaluated the frameworks. There is a lack of studies on the implementation of HP competencies. | Ireland and Malta (e) |
| Battel-Kirk, B. and Barry, M. M. (journal article) | Implementation of health promotion competencies in Ireland and Italy—A case study | 2019 | Case study including desk reviews and semi-structured interviews | Exploring the promoting and challenging factors for implementation of CompHP at a national level comparing two countries, Ireland and Italy | The progress of CompHP implementation reflected the HP infrastructure and capacity in the countries. Major limitations were a lack of awareness of the CompHP also by main stakeholders and employees. The CompHP has to be promoted over the next years to build capacity of HP. | Ireland and Malta (e) |
| Battel-Kirk, B. and Barry, M.M. (journal article) | Evaluating progress in the uptake and impact of health promotion competencies in Europe | 2020 | Online survey conducted with consortium included in the development of CompHP | Identify attitudes regarding the CompHP, level of current and intended use and opinions on their impact | Only 81 responses were received from 25 countries: attitudes were generally positive, while only 53% confirmed the use of the CompHP in their country. The competencies were mainly used in education of HP. Main barriers for implementation of the CompHP was the lacking recognition of key organizations and stakeholder at a national level. | Ireland and Malta (e) |

**Category: Quality**

| | | | | | | |
|---|---|---|---|---|---|---|
| Wright, M. T., Noweski, M. and Robertz-Grossmann, B. (journal article) | Quality development in primary prevention and health promotion. A survey of the | 2012 | Quantitative online survey | Identify implementation of quality development strategies and standards | The results are very limited. Responses confirmed a great variety of quality standards used | Germany (g) |

*(Continued on following page)*

**TABLE 3 |** (*Continued*) Data extraction of original publications identified in the scoping review chronologically ordered by category and language published (Professional characteristics of health promotion: a scoping review of German and international literature, Switzerland, 2021).

| Authors (nature of literature) | Title | Year | Method | Aim | Outcome | Country |
|---|---|---|---|---|---|---|
| | member organizations of the Federal Association for Prevention and Health Promotion in Germany | | | used in HP and prevention in Germany | in HP and prevention. Therefore, the questionnaire could not be answered properly. The authors concluded that a qualitative research approach is necessary to capture the variety of quality standards used in HP and prevention. | |
| Bär, G., Noweski, M., Ihm, M. and Voss, A. (journal article) | Prevention of overweight in children: Standard setting documents | 2016 | Literature review | Assessing quality standards of overweight prevention in children found in databases and google | German literature on quality standards and overweight prevention in children is rare. Standard setting documents of key stakeholders of HP and prevention in Germany are comparable with each other but do not refer to each other. Moreover, it is not clear which document guides the quality standards for overweight prevention in children in Germany. | Germany (g) |
| Grossmann, B. and Noweski, M. (journal article) | Quality in primary prevention. Results of a survey of members of the Federal Association for Prevention and Health Promotion | 2016 | Qualitative study using expert interviews | Exploring the current status of quality standards used in HP and prevention | 42 experts were interviewed: Using quality standards in practice correlates with financial resources and qualified staff in the institutions. Good practice examples would be a helpful tool as well as a nationwide monitoring system. First, more funding is required for assuring quality standards in practice. | Germany (g) |
| Reisig, V., Kuhn, J., Loos, S., Nennstiel-Ratzel, U., Wildner, M. and Caselmann, W. H. (journal article) | Primary prevention and health promotion in Bavaria: Taking stock | 2016 | Mixed methods approach including an online survey and expert interviews | Assessing the status quo of prevention and HP in Bavaria, aiming at a quality-oriented development of the field | HP and prevention practice mainly address health literacy and mental health issues by providing health information relating to behavior change interventions. Rather low is the engagement with gender specific topics or socially disadvantaged groups including working with a setting approach in communities. About half of the participants include scientific results in their project development and 43% conduct evaluations. | Germany (g) |
| Noweski, M., Bär, G., Voss, A., Ihm, M. and Fricke, L. (journal article) | Common quality standards in primary prevention. The expert survey PräKiT | 2018 | Qualitative study using expert interviews | Identifying the need of common quality standards for HP and prevention | A common quality standard is meaningful to the interviewed experts. A complex and challenging process to identify this common standard is expected to meet specific needs of different topics within HP and prevention. Even within these expert interviews no theoretical saturation could be revealed because expectation varied meaningfully. | Germany (g) |

[a]HP: health promotion
[b](g): German.
[c](e): English.

HP interventions [48]. Further publications accentuate the importance of the implementation of ethics into HP practice [49,50] and also national guidelines for HP [51]. It is evident that there is a substantial gap between theory and practice regarding HP ethics, despite ethical considerations being mandatory when planning and conducting interventions. As already recognized within the category "profession," a code of ethics in HP is needed and asked for [41,46–48,50,51].

## Education/Training

In total, nine original publications were identified that fit the category "education/training." German as well as English publications display a broad spectrum of thematic aspects. Some publications evaluate professional training programs in HP for different professions [52–56]. These publications demonstrate that persons employed in HP often don't have any professional knowledge about it [55]. There are also publications that study the HP competencies within a master's program in Public Health [32,57], which reveal major gaps in the curriculum addressing social determinants of health [32]. Finally, only a few publications focused on academic study programs in HP. Only two were found from German speaking regions [58,59] and these studies are not exclusively focusing on HP but also included related study topics courses.

## Competencies

Competencies are the basis for professional performance. In total, nine publications were identified focusing on competencies in HP, two of them in German. The German publications don't focus on the CompHP [60,61] but developed an inclusive framework for PH, health sciences and HP study programs [59]. Most English publications focus on the development and implementation of the CompHP [6,62–64], which is also adapted as an accreditation framework for HP practitioners [65]. Australia [66] and New Zealand [67] also edited own frameworks. The importance of common professional competencies is affirmed by all authors [6,60–66]. It is seen as the basis for a common professional profile, for transparent professional competencies, for a common code of ethics and strong professional identities as HP practitioners [63]. For more detailed literature on the competencies, Battel-Kirk's and Barry's scoping review from 2019 can be consulted [63].

## Quality

All five original publications focusing on quality assurance of HP and prevention interventions included in the content analysis of this scoping review were conducted in Germany. Their overall aim was to identify quality standards that can be used by primary stakeholders in HP and prevention [68–71]. Quality assurance of HP and prevention interventions are still fragmented and poorly standardized, although a standardization of tools and activities is agreed upon when resources are available. Regarding the quality of HP interventions as defined in the Ottawa Charter, one article detected an insufficient focus on social determinants of health or on projects based on the setting approach but a strong focus on behavior change programs [7].

## DISCUSSION

This scoping review aimed to outline the actual developments of professional characteristics of HP with a focus on the German speaking part of Europe taking international literature into account. Of 90 publications that were identified, 37 were original publications and about one-third of the theoretical and original publications were contributions from the German speaking context. Based on the content-analysis, the results were summarized in categories that reflect the professional characteristics: 1) profession, 2) ethics, 3) education/training 4) competencies, and 5) quality. Content analysis was only applied to the original literature ($n$ = 37). Overall, HP is clearly developing as a profession especially in English speaking countries, as literature was identified regarding all professional characteristics [16] with the exception of the professional identity of HP practitioners. German original literature did not address "ethics" as a professional characteristic and only few publications considering "competencies" and "education/training" were identified. Thus, the discussion in the German original literature was mostly focused on the two specific professional characteristics of "education/training" and "quality." Furthermore, the quality of the German literature was lower than the international literature (i.e., more grey literature than peer-reviewed articles). Summing up the findings, the importance of a common code of ethics in HP, a meta theory in HP and a common professional profile including professional competencies are needed to promote professionalization of HP regarding professional identity and professional performance of HP practitioners.

In total, German original literature revealed that HP is rather unclear in its conception and establishment as a profession, as well as in terms of its relation to PH. The following factors contribute to the confusion and intransparency of HP at an educational level, within the labor market and in society. As described in the introduction HP is not mentioned to be an exclusive profession but is seen as a paradigm shift of our health system [1]. Therefore, on the one hand a wide range of professions (doctors, nurses, therapists, teachers, social workers etc.) are called to promote health and integrate the principles of the Ottawa Charta in their professional work: participation, empowerment, and health equity. On the other hand, we need HP professionals to initiate this paradigm shift, conduct specific HP projects and mediate between important stakeholder to foster Health in all Policies [12,34]. This can be contradictory and in conflict with the goal to define clear professional roles. There is a lack of studies describing the contents and learning outcomes of HP programs, which are needed to promote comparability and transparency of the programs to define the professional roles of HP practitioners. The confusion about health-related courses is reinforced by a great expansion of these programs since the Bologna process in 1999. Study programs are named very differently (health sciences, health communication, public health, health management, health promotion etc.) but competencies and curricula are overlapping [26,72,73] and seem to qualify students for HP practice. Educational institutions competitively advertise for potential

**TABLE 4 |** Descriptive analysis of data included in the scoping review (Professional characteristics of health promotion: a scoping review of German and international literature, Switzerland, 2021).

| | Total | Language | |
|---|---|---|---|
| | N | German | English |
| | 90 (100%) | 31 (34.4%) | 59 (65.6%) |
| **Form of literature** | | | |
| Theoretical | 53 (58.9%) | 18 (20.0%) | 35 (38.9%) |
| Original article | 37 (41.1%) | 13 (14.4%) | 24 (26.7%) |
| **Categories** | | | |
| Profession | 34 (37.8%) | 10 (11.1%) | 24 (26.7%) |
| Theoretical | 26 | 7 | 19 |
| Original article | 8 | 3 | 5 |
| Ethics | 21 (23.3%) | 5 (5.6%) | 16 (17.8%) |
| Theoretical | 15 | 5 | 10 |
| Original article | 6 | — | 6 |
| Education/training | 16 (17.8%) | 6 (6.7%) | 10 (11.1%) |
| Theoretical | 7 | 3 | 4 |
| Original article | 9 | 3 | 6 |
| Competencies | 12 (13.3%) | 4 (4.4%) | 8 (8.9%) |
| Theoretical | 3 | 2 | 1 |
| Original article | 9 | 2 | 7 |
| Quality | 7 (7.8%) | 6 (6.7%) | 1 (1.1%) |
| Theoretical | 2 | 1 | 1 |
| Original article | 5 | 5 | — |

students with a great variety of study programs which is fostered since the Bologna process. Mainstreaming of HP can be recognized in many health-related study courses. This mainstreaming of HP in diverse sectors of health [2,22,74] may lead to loss of quality of HP practice. To establish an efficient PH system including a strong HP workforce, collaboration with a common language has to be the goal instead of competing interests and economic factors at educational institutions. More specific HP programs must be established at universities with staff engaged in HP research and practice. PH has to recognize the specific competencies for HP, which are not sufficiently addressed in general PH programs, even less in other health professions' programs [34]. Literature identified in the scoping review conclude that higher education institutions are meant to play a significant role to clarify HP's roles, promote its societal status and coordinate initiatives to overcome the mainstreaming of HP [12,14,41].

Concerns about the quality of HP and prevention in practice were identified in German original literature in terms of the poor standards in planning, conducting and evaluating HP interventions [7,68–71]. Most interventions are not based on a setting approach and do not focus on social determinants of health at a community level but address individual behavior change instead [7]. Of course, the debate about the quality in HP practice is inherently linked with competencies and the education or training of persons working in the field. Both, the German and English literature point out that there is a lack of expertise and professional knowledge in HP practice

[6,7,14,42,45]. Therefore, the CompHP framework by the IUHPE is a good basis to build on, for educational purposes, for the labor market to prioritize professional profiles, as well as for governmental guidelines for HP [37]. The international literature reveals few publications on competencies in HP, mostly the CompHP, but the implementation in the different countries is progressing slowly [6,64]. The German version of the CompHP was published in 2014 [75]. Whereas original literature showed no reference to the CompHP in 2015 [60], theoretical publications in 2018 indicate the rising recognition of the CompHP also in the German speaking part of Europe [75,76]. Battel-Kirk and Barry (2019) point out that the implementation progress of the CompHP reflects the professionalization of HP in the particular country [6,64]. The competency framework for HP is meant to strengthen professional identity and roles of the HP workforce [63]. Major challenges for the implementation are structural aspects, e.g. political commitment to HP, lacking differentiation between HP and PH, a weak HP workforce and lacking knowledge about the CompHP. Facilitators are a strong HP professional institution, national accreditation of HP and HP education [6,63]. Therefore, the CompHP influences all professional characteristics of HP, namely professional performance and professional identity, ethics/values, education/training and competencies/quality. Two further initiatives are mentioned which promote clarification regarding the professional characteristics of HP: Quality of HP practice in the German speaking countries is enhanced by "Quintessenz" a widely known online network and project-management-tool, which ensures a systematic and high quality proceeding in HP practice [77]. An internationally committed code of ethics in HP still needs to be established. Besides the CompHP the code of ethics of health education professionals in the United States [78] can also be drawn upon. There have been made great professional achievements in HP within the last years (see **Table 1**). Therefore, we need a young professionally trained HP workforce to further promote their visibility and competencies to address actual and future challenges of society's health.

## Limitations
Searching the databases and journals revealed an abundance of English literature but a scarcity of German literature. Therefore, we conducted an extended search for grey literature in Google Scholar and a hand search for the German literature. This may have led to an increased amount of grey literature in the German relative to the English literature. As the main aim of the scoping review was to identify professional characteristics of HP in the German speaking part of Europe with comparison to the international literature, this limitation should not invalidate the findings. Further, abstracts and full texts were only single-screened, which represents a methodological limitation. However, this proceeding is often applied in scoping reviews and can even be justified for systematic reviews [79].

## Conclusion

Analyzing international literature on professional characteristics of HP reveals the necessity of clarifying competencies and the professional profile of HP. The lack of expertise and professional knowledge in HP practice in the German speaking part of Europe as well as internationally [6,7,14,41,44] indicates the need for clarification regarding specialized competencies of HP within the PH field. Therefore, integrating the CompHP in national and regional HP institutions, at an educational level, as well as in relation to the workforce and labor market would help to address these needs. The lack of recognition of the CompHP is evident in the identified literature, especially in the German speaking countries, and must be of concern in future investigations on the HP profession. The unclear relationship and differentiation between HP and PH further hinders the conceptualization of the HP profession [6,62]. Higher education institutions play a major role in clarifying the interconnectedness between HP and PH, addressing this by adapting curricula and names of their study programs. HP as a profession and HP professionals would benefit from being trained based on the CompHP, which reflects professional competencies for HP based on the Ottawa Charta. Transparency and comparability of competencies of the HP, PH and the wider health workforce contribute to the political and societal recognition of important professions in national and international efforts to tackle rising social inequalities of health and global health issues (infectious and non-communicable diseases).

## AUTHOR CONTRIBUTIONS

VB developed the conceptual idea, conducted the scoping review and wrote the manuscript. TG supervised the project and contributed to the manuscript. FW contributed substantially to the conceptual idea of the project and the manuscript and supervised the project.

## CONFLICT OF INTEREST

The authors declare that the research was conducted in the absence of any commercial or financial relationships that could be construed as a potential conflict of interest.

## ACKNOWLEDGMENTS

We would like to thank Victoria Saint for proof reading and editing the English of the manuscript.

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
