## [Reviewer comments · International Journal of Public Health]

Peer Review Report

Review Report on Professional characteristics of health promotion: a scoping review of the German and international literature

Review, Int J Public Health

Reviewer: Dean Whitehead

Submitted on: 12 Apr 2021

Article DOI: 10.3389/ijph.2021.1603993

EVALUATION

Q 1 Please summarize the main theme of the review.

The main theme is relating the context of health promotion activities, aligned to health professionals, in capturing the German-specific literature as it relates to the wider international-related literature.

Q 2 Please highlight the limitations and strengths.

The strength of this review is in its mostly systematic scoping process - and, particularly, in identifying the 'nuances' and limitations of finding the German-specific literature i.e. in filtering through the grey literature sources. The main limitations are a minor methodological error - and the 'analysis' section.

Q 3 Please provide your detailed review report to the authors, structured in major and minor comments.

Overall, this is a well structured and well articulated review. As already hinted at - there are just two areas that I would like to see amended:

1. The PRISMA process highlights the separation of quantitative and qualitative studies (for the purpose of appraisal) - but does not identify that there are a number of mixed methods studies involved. Mixed methods is its own paradigm.
2. The 'data' analysis/synthesis/appraisal process is 'missing'. The authors state that the detail is included in table 3 - but this is not the case. It is just a percentage breakdown of the emerging themes. Any critical appraisal/synthesis process should be reported more fully. Did the authors use any tools i.e. CASP?
3. The 'themes' do not appear that illuminative or descriptive' - and more as brief terms. It would be good to have more detail on how they emerged and how they are representative.

PLEASE COMMENT

Q 4 Does the reference list cover the relevant literature adequately and in an unbiased manner?

Yes

Q 5 Does this manuscript refer only to published data? (unpublished data is not allowed for Reviews)

Yes.

Q 6 Does the manuscript cover the issue in an objective and analytical manner

Yes.

Q 7 Was a review on the issue published in the past 12 months?

No.

Q 8 Does the review have international or global implications?

Yes - it highlights what is already known about the international context - but compares to the German context - which it reports may actually 'lag behind' much of the wider international literature findings.

Q 9 Is the title appropriate, concise, attractive?

Yes - straight to the point

Q 10 Are the keywords appropriate?

Yes

Q 11 Is the English language of sufficient quality?

Yes - I am a native English speaker - and see nothing of concern - just the 'usual' expected off typos, grammatical errors that could be addressed at proofing stages.

Q 12 Is the quality of the figures and tables satisfactory?

Yes.

QUALITY ASSESSMENT

Q 13 Quality of generalization and summary

Q 14 Significance to the field

Q 15 Interest to a general audience

Q 16 Quality of the writing

REVISION LEVEL

Q 17 Please take a decision based on your comments:

Minor revisions.